# Scalable SCADA-driven Failure Prediction for Offshore Wind Turbines Using Autoencoder-Based NBM and Fleet-Median Filtering

Ivo Vervlimmeren<sup>1,2,4</sup>, Xavier Chesterman<sup>1,2,3,4</sup>, Timothy Verstraeten<sup>1,2,4</sup>, Ann Nowé<sup>2</sup>, and Jan Helsen<sup>1,3,4</sup>

**Correspondence:** Ivo Vervlimmeren (ivo.vervlimmeren@vub.be)

Abstract. Offshore wind turbines are crucial for sustainable energy production but face significant challenges in operational reliability and maintenance costs. In particular, the scalability and practicality of failure detection systems are a key challenge in large-scale wind farms. This paper presents a scalable, comprehensive approach to failure prediction based on the Normal Behavior Modeling (NBM) framework that integrates three components: a cloud-based pipeline, an undercomplete autoencoder for temperature-based anomaly detection, and a time-aware anomaly filtering method. The pipeline enables dynamic scaling and streamlined deployment across multiple wind farms. The autoencoder was trained exclusively on healthy 10-minute SCADA data and produces detailed anomaly scores that serve as the input for our filtering technique. It was trained on four years of data from a large offshore wind farm in the Dutch-Belgian zone and achieved UHH-ratios (UnHealthy-Healthy) of up to 1.69 and 1.21 for the generator and gearbox models, respectively. The filtering method refines the raw anomaly scores by comparing turbine signals to a windowed fleet median. By aggregating scores via sliding windows and employing robust distance metrics, the method reduces the volume of anomaly scores by up to 65% without sacrificing predictive accuracy. This selective filtering effectively minimizes noise and non-relevant anomalies, enhancing the efficiency of maintenance analysis.

# 1 Introduction

Global initiatives to mitigate climate change and the need for sustainable energy production are driving the increasing focus on renewable energy sources (Gielen et al., 2019). Offshore wind turbines, in particular, play an essential role in this transition due to their ability to harness the strong and relatively consistent wind resources available at sea. However, the operational reliability of these turbines is crucial for maintaining the economic viability of wind farms (Dao et al., 2019). Unplanned maintenance and downtime contribute significantly to overall operating costs for offshore wind farms, accounting for up to 30% of the Levelized Cost of Energy (LCoE) (Stehly et al., 2020). Consequently, improving the maintenance strategies is essential for reducing costs and enhancing the economic sustainability of offshore wind energy.

<sup>&</sup>lt;sup>1</sup>AVRG, Vrije Universiteit Brussel, Pleinlaan 3, 1050 Brussels, Belgium

<sup>&</sup>lt;sup>2</sup>Artificial Intelligence Lab, Vrije Universiteit Brussel, Pleinlaan 9, 3rd floor, 1050 Brussels, Belgium

<sup>&</sup>lt;sup>3</sup>Flanders Make@VUB, Pleinlaan 2, 1050 Brussels, Belgium

<sup>&</sup>lt;sup>4</sup>OWI, Vrije Universiteit Brussel, Pleinlaan 2, 1050 Brussels, Belgium

Advancements in turbine design and the increasing availability of sensor data have made condition monitoring systems an effective component of offshore wind turbine maintenance (Helsen, 2021). Supervisory Control and Data Acquisition (SCADA) systems, in particular, enable the continuous monitoring of turbine performance, facilitating the early detection of faults. This proactive approach enables predictive maintenance, allowing operators to implement strategies that reduce the likelihood of catastrophic failures, optimize maintenance schedules, and minimize associated costs. One prominent approach in this domain is Normal Behavior Modeling (NBM), which uses historical data to establish models of expected system behavior, and flags deviations from the predicted behavior as potential anomalies (Chesterman et al., 2023).

However, the scalability and practicality of failure detection systems remain key challenges in large-scale wind farms. As more offshore wind farms come online to support the growing demand for renewable energy (Díaz and Soares, 2020), the sensor data generated by monitoring systems increases proportionally. Processing this data efficiently is crucial to ensuring timely and accurate fault detection. An answer to this issue is the development of scalable and modular systems, which are particularly advantageous, as they enable seamless integration of new turbines and data streams while maintaining adaptability through hyperparameter optimization or manual configuration changes. Such systems can also enable continuous improvement in predictive models through methods such as fine-tuning and transfer learning. However, the question of how to process the growing amount of data is not the only issue. The increase in input data inherently leads to a proportional growth in the volume of results. In the context of failure prediction using Normal Behavior Modeling (NBM), this results in the generation of a prediction for each relevant signal and each turbine. Given the many turbine components that need to be examined regularly to facilitate predictive maintenance practices, analyzing the resulting large amount of predictions is challenging. This underscores the necessity for automated post-processing and filtering mechanisms to reduce noise, and systematically highlight meaningful and useful observations.

This paper investigates an approach that compiles multiple predictions of anomaly detection methodologies and combines them into more reliable outcomes. Specifically, we propose a scalable methodology that integrates state-of-the-art SCADA-based anomaly detection using temperature signals with a statistical approach to filter the resulting anomaly scores to decrease false positives. We start by introducing a temperature-based NBM anomaly detector utilizing an autoencoder model trained on 10-minute SCADA data. We then develop a robust and scalable pipeline capable of using this detector to process multiple wind farms with diverse parametrizations, enabling extensive performance evaluation. Finally, we propose a computationally efficient, time-aware filtering technique that employs the fleet median as a reference point to remove non-relevant anomalies, enabling better, faster, and more automated alarming.

We start with an implementation of the NBM framework; we chose this method due to its widespread use and proven capability (Chesterman et al., 2023). For the model, we used an autoencoder trained exclusively on healthy data. To give the model more robustness against anomalous data (unhealthy data) that might not have been filtered from the healthy data during preprocessing, an undercomplete architecture (which means that the latent space is of a lower dimensionality than the input/output space) is used.

With this anomaly detector as a base, we turn toward the scalability problem, both in training, using, and fine-tuning. Our answer to this issue is a scalable pipeline that can quasi-automatically manage our autoencoder-based NBM or any other

NBM, as long as it adheres to the general NBM framework. This pipeline architecture supports cloud-based implementation, facilitating ease of deployment and dynamic scaling to ensure optimal performance under diverse operational conditions. This helps during training and inference, enabling us to easily change configurations, experiment with different models and parameters for different farms and machines, and run automatic hyperparameter optimization. As it is cloud-based, horizontal scaling is straightforward, allowing additional instances of the entire pipeline to be deployed as needed. We used this pipeline to train and process data from two farms, investigating different models and parameters.

Finally, we examine the output of the NBM. Our autoencoder-based NBM was applied to 10-minute SCADA data collected over approximately four years from a large wind farm in the Dutch-Belgian offshore zone, which has experienced several gearbox and generator failures. This resulted in a substantial number of anomaly scores: with 16 predicted signals per turbine, we generated over 320 1-hour time series spanning four years. Since most anomalies are irrelevant when predicting generator or gearbox failures (due to model noise, natural weather-related variations, or their lack of association with the component failures of interest), we desire the ability to remove these non-relevant anomalies to facilitate and enhance analysis. To do this, we make the assumption that at any given time, most turbines in the fleet exhibit normal behavior. To define this, we use the raw anomaly scores to construct a windowed, multidimensional fleet median per signal as a representation of normal behavior. We infer from this that turbines whose signal remains close to this fleet median for a certain time window can be considered as exhibiting normal behavior, allowing us to discard the corresponding anomaly scores for that time window. We evaluate several combinations of distance calculations and thresholding techniques and find that we can filter out up to 65% of the raw anomaly scores while the adjusted scores still retain equivalent predictive power. This significantly accelerates both manual and automated analysis, thereby improving the efficiency of alarming. In addition, since this method is purely subtractive, the likelihood of a false positive is, at worst, the same as with the original anomaly scores.

In summary, this work aims to enhance our ability to perform failure prediction at scale by developing a scalable pipeline for NBM failure detection and implementing efficient anomaly score filtering techniques, which improves the operational reliability and operating costs of offshore wind farms and ultimately contributes to the broader goal of sustainable and cost-effective renewable energy production.

The organization of this paper is as follows. We briefly examine some related work in the following section (Sect. 2); in the subsequent Methodology section (Sect. 3), we begin with the anomaly detector (Sect. 3.1) by first introducing the general NBM framework followed by a description of our autoencoder-based implementation. After this, we present the scalable pipeline (Sect. 3.2) before describing the various methods employed in the filtering of anomalies (Sect. 3.3). In the result section (Sect. 4) we show the performance of our autoencoder-based NBM (Sect. 4.3) and anomaly filtering method (Sect. 4.4). Finally, we present our conclusions and discuss potential future directions in Sect. 5.

# 2 Related work


The growing need to optimize the operational reliability of offshore wind turbines has led to significant advancements in predictive maintenance, condition monitoring, and anomaly detection. Early work in this area focused on Supervisory Control

and Data Acquisition (SCADA) systems (Yang et al., 2013), which allows for continuous monitoring of turbine performance and the early detection of failures. SCADA-based condition monitoring has been a fast-growing field (Tautz-Weinert and Watson, 2017b), (Maldonado-Correa et al., 2020) and, failure prediction, specifically, has developed several approaches (Black et al., 2021).

One of the most prominent approaches is Normal Behavior Modeling, where models of expected system behavior are created using historical data, and deviations from this behavior are labeled as potential anomalies. Even within this category, a glut of techniques is currently being investigated (Chesterman et al., 2023). NBM has shown promising results in various studies related to wind turbine monitoring. For instance, NBM with Artificial Neural Networks has been demonstrated to achieve notably high accuracy in detecting deviation from the nominal power curve (Ciulla et al., 2019). Similarly, work by Wei et al. (2019) highlights the role of NBM in enhancing turbine health monitoring by using a combination of SCADA data and advanced machine learning models. Machine learning models like autoencoders have proven particularly suitable for failure prediction in wind turbines (Lee et al., 2024; Liu et al., 2023; Miele et al., 2022; Chen et al., 2021; Renström et al., 2020; Beretta et al., 2020). Implementing solutions for handling the vast pool of data generated by sensor networks and SCADA systems comes with several challenges, like gathering a substantial amount of data of often disparate origin into a combined, usable, dataset (Helsen et al., 2016). Said data needs to fulfill certain requirements, i.e., volume, velocity, variety, etc (Nabati and Thoben, 2017). But such systems can be quite successful, as shown by Canizo et al. (2017), who created a robust big data processing framework, predicting the status of 100 wind turbines with 80% accuracy based on historical data. Though their work primarily focused on a static deployment of the big data failure prediction system for a single farm, whereas our approach additionally enables rapid prototyping and flexible configuration adjustments.




With regards to our postprocessing method, the idea that the increased amount of data from more turbines can also be leveraged directly for failure detection or enhancement thereof is known. This is usually done by comparing to some central tendency of the fleet, Hendrickx et al. (2020b) clusters raw sensor data from a fleet of machines using a distance measure based on the amount of warping in Dynamic Time Warping. While Li and Wu (2020) also uses raw sensor data to generate a vector containing the difference with the fleet median for every turbine, after which they use a vector autoregressive model and VAR control charts to detect anomalous behavior. Similarly, filtering the anomaly scores is also known, as shown by Li et al. (2020), who extracted and clustered known false positive anomaly sequences to calculate exemplars, which they then used as a measuring stick for other anomaly sequences. However, they do not use the fleet's central tendency in their method. The novelty of our postprocessing method lies in its general, uncomplicated, and efficient filtering technique, which utilizes the anomaly scores themselves as input and then also leverages the fleet's central tendency (in this case, the median) to detect and discard non-relevant anomalies. It is specifically aimed at streamlining the processing of anomaly scores, allowing for faster analysis and improving the accuracy and effectiveness of the automated alerting system.

# 120 3 Methodology


In this section, we briefly discuss the general characteristics of the Normal Behavior Modelling framework. Then, we introduce an implementation that uses SCADA temperature signals to drive an autoencoder model to obtain an anomaly score. After this, we will detail how we created and deployed a scalable pipeline to manage and deploy any NBM implementation for failure detection. Finally, we present a versatile method to filter out failure-unrelated anomalies from the excess of anomaly scores produced by large-scale anomaly detection.

# 3.1 Anomaly detection

#### 3.1.1 The Normal Behavior Model Framework

The Normal Behavior Model (NBM) framework is widely utilized in wind turbine condition monitoring and focuses explicitly on predictive maintenance. Such an NBM is trained on historical sensor data and subsequently attempts to predict specific signals given several essential signals like active power, windspeed, etc. The error in its prediction is analyzed, and if it is deemed significant, it will be classified as an anomaly. There are myriad ways to implement this, but most normal behavior models will use a roughly similar structure.

Figure 1. NBM framework overview as shown in Chesterman et al. (2023)

An NBM needs a sizable collection of historical data from sensors installed on one or more wind turbines. These sensors monitor various operational parameters, including temperature, vibration, power output, and wind speed. This data then generature,

ally needs some level of preprocessing based on its properties and the specific requirements of the NBM. Typically, noise and absurd outliers are removed, missing values are handled, and the data may be scaled, normalized, or even transformed. This process tends to be crucial for the robustness and accuracy of the model.

Then, the cleaned data is split and used to train the ML model that attempts to predict signal values observed during normal and healthy operational conditions. There are many possible algorithms for this, which can generally be sorted into three categories: statistical models, shallow machine learning, and deep learning. In many cases, an ensemble of multiple models will be used.

Once trained, the model's predictions can be used to calculate the prediction error, i.e., the difference between the observed and predicted values. Again, there are several methods for this, like statistical and machine learning, each with its own benefits and drawbacks (Chesterman et al., 2023).

# 145 3.1.2 Data preprocessing





The data that is used in this paper comes from real operational wind farms. This means that its observations are not labeled as healthy or unhealthy. In principle, an autoencoder model does not need healthy training data. Undercomplete autoencoders can learn themselves what the most relevant dimensions of the problem are. In the case of anomaly detection, this means that it, in principle, can learn the normal behavior. Nevertheless, it might improve the performance of the model if only healthy data is given during the training phase.

The procedure to select healthy data is based on two data sources. Firstly, a failure list created by the wind turbine operator containing several major failures. Secondly, a list of forced shutdowns was extracted from the status logs of the wind turbine. All data that precedes the major failures by less than 4 (WF1) or 6 months (WF2), or follows it by less than 180 days is considered to be unhealthy. For the forced shutdowns, all data that precedes or follows such a shutdown by less than 6 hours is also considered unhealthy.

To train and test the autoencoders properly, the data is divided into three datasets, i.e. a training, validation, and testing dataset. The training dataset consists of the first (chronologically) 3000 healthy observations for each wind turbine. This guarantees that the training dataset has an equal amount of healthy data for each wind farm. The validation dataset consists of the next 2500 healthy observations for each wind turbine. These two datasets are used during the hyperparameter tuning. The remaining data is assigned to the testing dataset.

The autoencoder uses data that is aggregated to the 1-hour level as input. The decision to aggregate is based on several considerations. Firstly, the failures that are the focus of this research are issues that form relatively slowly over time (several days to several months). This does not require data that has a 10-minute resolution. Secondly, by aggregating the data, the amount of noise can be reduced. Thirdly, the aggregated dataset is smaller in size, and processing it requires less computational power.

Data analysis showed that the data contains a small but not negligible amount of measurement errors. These errors are often quite obvious. Strongly negative component temperatures (lower than the ambient temperature) are impossible to explain physically and can be assumed to be measurement or sensor errors. A second way the measurement errors show themselves

is as unrealistically high temperature readings. These have the value 205 °C, which seems to be a maximum or default value.

Both the extremely low and suspicious high values are replaced by missing values. This is done by defining for each signal upper and lower bounds.

Missing values are removed instead of imputed. The analysis of the missingness indicated that the missing data is most likely Missing Completely At Random (MCAR) or Missing At Random (MAR). This means that the probability that removing the observations with missing values introduces bias is small.

$$x_{norm} = \frac{x - min(x_{train})}{max(x_{train}) - min(x_{train})}$$
 (1)

As a final step, the data is normalized (which is not the same as standardizing). This is done by calculating the minimum and maximum of each signal on (only) the training dataset. The min is subtracted from the signal value and the difference is divided by the difference between the maximum and the minimum of the signal (see Eq. 1).

# 3.1.3 The autoencoder-based NBM




To model the normal behavior of a wind turbine many supervised and unsupervised statistical, machine learning and deep learning algorithms can be used (Tautz-Weinert and Watson, 2017a; Black et al., 2021; Chesterman et al., 2023). All these algorithms have advantages and disadvantages. Statistical algorithms tend to be more data efficient and less computationally heavy than deep learning algorithms, but the latter are in general much more capable modelers of non-linear relations. Which algorithm to choose depends on the properties of the problem that is being modelled. The complexity of the problem, the data availability, the computational limits, ..., are all factors that need to be taken into account.

In the course of several years of research that focused specifically on the wind turbine context, which was published in Chesterman et al. (2023) and Chesterman (2024), experiments were carried out with different statistical, machine learning and deep learning algorithms. Although statistical and shallow ML learning solutions are less computationally demanding, the autoencoder algorithm was identified as the most suited for the problem at hand. This has several reasons. Firstly, the modeling accuracy of the autoencoders on the wind turbine data is in general superior to that of the other tried methods. Secondly, the fact that extensive hyperparameter tuning indicates that a shallow configuration of the autoencoders is optimal, keeps the computational demand of training the algorithms low. Thirdly, the fact that autoencoders practically out of the box can be used in multi-input multi-output (MIMO) settings, gives them an extra computational advantage. These advantages tipped the balance in their favor, even though they also have disadvantages. Table 1 in Sec. 4.2 shows the computational efficiency that can be achieved with a well-tuned autoencoder and a careful, minimal selection of data compared to training an ensemble of statistical models.

Autoencoders are deep learning algorithms that consist of two parts, i.e. an encoder (a function f) and a decoder (a function g). The goal of the encoder is to map the original data to a latent space with lower (undercomplete autoencoder) or higher (overcomplete autoencoder) dimensionality (h = f(x)). The decoder uses this representation (h) to recreate x. The autoencoder is given a set of inputs x, and its goal is to reproduce x as well as possible. This is done by minimizing a loss function

L(x,g(f(x))). The loss function penalizes g(f(x)) for being different from x. An example of such a function is the MSE (Mean Squared Error) (Goodfellow et al., 2016). If anomaly detection is the goal undercomplete autoencoders are particularly useful because they tend to ignore rare events like anomalies. The reconstruction error L, which is the difference between the observed and reconstructed values, can then be searched for abnormal deviations or patterns.

The autoencoder is set up as a MIMO model. This implies that the normal behavior of several signals is modeled at the same time. Figure 2 gives a schematic representation of how this is done. The autoencoder has in its input layer a number of neurons that is equal to the number of input signals. The output layer has the same number of neurons. This means that in a single run, the normal behavior for all the signals is predicted. This is a considerable advantage since it reduces the run-time of the algorithm. The reconstruction error is then the difference between the original signal values and the predicted values.

Figure 2. Schematic representation of an undercomplete autoencoder


To determine the optimal values for the different hyperparameters, i.e. number of neurons per layer, number of hidden layers, activation functions, and learning rate of the model, the Keras Hyperband tuner is used. The search space is however limited to undercomplete autoencoders only.

# 3.2 A Cloud-Based, Scalable Pipeline for Anomaly Detection

This section details the development of a scalable pipeline designed to efficiently deploy various anomaly detection models and facilitate continuous, pseudo-real-time analysis for predictive maintenance. By leveraging a robust anomaly detector as a foundation, this pipeline addresses scalability challenges in training and fine-tuning, allowing for seamless management of autoencoder-based NBM models and other models within the general NBM framework. Its cloud-based architecture supports

dynamic scaling and easy deployment. It enables experimentation with different configurations and makes it easier to tailor models to specific operational conditions, leading to more reliable turbine operation analysis.

Figure 3. Scalable Pipeline architecture as deployed by us





The standard pipeline architecture (Fig. 3) comprises four core modules: data cleaning, data preprocessing, anomaly detection, and failure diagnosis, mirroring the standard NBM framework. These containerized modules are configured into a modular pipeline that supports dynamic orchestration, which enables horizontal scaling and allows it to handle spiking or custom computational demands during training or inference. This has been accomplished through technologies such as Docker and Kubernetes, which are used for automatic container deployment and resource allocation. The pipeline can also interface with distributed storage solutions, like Hadoop, to efficiently store and manage the data it requires and produces. The pipeline can be run locally, as pure prototype code, or on a local docker, supporting efficient prototyping in various environments. It provides functionality to automatically package module code and upload a container image to a repository, from where it can be deployed on cloud resources.

The cloud-based design and containerization allow individual modules, or even the entire pipeline, to be scaled independently, enabling rapid adaptation to varying data loads or model requirements. For example, deploying a differently configured pipeline instance for a new wind farm that requires one or more custom data preprocessing modules. Dynamic scaling is further supported by the architecture's ability to allocate additional computational resources for parallel processing at the request of individual modules, allowing machine learning models to leverage distributed computing when appropriate and/or possible. However, the model implementation must be adapted to leverage this feature of the pipeline framework.

The pipeline's modularity encourages adaptability, allowing individual components to be easily replaced or upgraded without affecting the overall system. This facilitates fast prototyping experimentation with different preprocessing methods, anomaly detection algorithms, or failure diagnosis approaches. Module instances can be shared; two different anomaly detector implementations could share the same data cleaning and failure diagnosis module instances. The flexibility also remains once the pipeline is deployed, and adding and removing extra modules is straightforward, e.g., enabling the addition of a root-cause

finder module. Such an architecture also aids hyperparameter optimization, targeting components separately or the pipeline as a whole and exploring the parameter space in a distributed fashion to take advantage of dynamic resource allocation. All these features enable the system to adapt dynamically to the specific challenges of different wind farms or turbine models.

Another benefit of the cloud-based architecture is that it allows for online learning and fine-tuning. Running continuously and ingesting fresh data, processing it efficiently without reprocessing older data wherever possible, and updating the predictive model it uses for inferencing. This can ensure that the predictions remain accurate and effective despite evolving operational conditions. Moreover, this architecture is robust and provides high availability and fault tolerance, reducing downtime and improving reliability. Important considerations for real-world deployments.

In summary, the scalable, cloud-based pipeline leverages modularity, containerization, and dynamic resource allocation to efficiently manage the training, inference, and fine-tuning of implementations of the NBM framework. Its adaptability to varying workloads, support for online learning, and ease of component integration make it an effective solution to streamline anomaly detection in offshore wind turbines. By combining robustness, flexibility, and efficiency, this architecture provides a reliable foundation for advancing operational monitoring and decision-making in real-world deployments.

# 3.3 Filtering Anomalies






In this section, we detail and explain the functioning of our anomaly filtering method and integrated sub-methods, which we compare in Sect. 4.4. This filtering method aims to significantly accelerate both manual and automated analysis by removing irrelevant (to the failures we wish to predict) anomalies without impacting the predictive accuracy of the signals, thereby improving the effectiveness of alarming. This section starts by laying out the intuition before describing the trivial one-dimensional implementation, which we then extend to the multi-dimensional case. After this, we present the distance calculation and thresholding methods before explaining our validation method.

Anomaly detectors typically detect substantially more anomalies than those only related to failure. They suffer from noise, false positives, false negatives, and non-relevant detections. These issues may have several causes, such as detector accuracy, edge cases, poorly reported maintenance, etc. This is not a significant problem for a limited number of machines, but analyzing the results for multiple large fleets becomes time-consuming. Employing robust post-processing methods to reduce noise and enable better and faster alarming immediately provides added value.

Also, anomaly detection at scale for wind turbines means monitoring a fleet, enabling us to leverage fleet-level knowledge as an additional factor in detecting anomalies. Sometimes, this can be explicitly incorporated during the anomaly detection itself (Hendrickx et al., 2020b). Additionally, various implementations of anomaly detection use the fleet median when preprocessing the data for normalization purposes (Chesterman et al., 2023). However, this can be problematic with some anomaly detection methods.

Since anomaly scores generally have an inbuilt severity, where higher scores represent more severe anomalies, a straightforward approach to filtering would be to discard low-level anomalies while retaining only the severe ones. However, this strategy risks overlooking significant patterns, such as an increasing frequency of lower-level anomalies, which might indicate

the gradual onset of more serious issues and lend more weight to newly appeared higher-level anomalies. Such trends could be valuable in informing swifter alarming mechanisms.

Our method addresses this by filtering the anomaly scores based on the entire fleet's information while incorporating a temporal aspect to at least partially capture these evolving patterns. Since this takes place after the actual anomaly detection and uses the produced anomaly scores, the method is agnostic towards the anomaly detection method used. Furthermore, since this method is purely subtractive, the likelihood of a false positive is, at worst, the same as with the original anomaly scores.

The core assumption of the method is that if the fleet is large enough, the median anomaly score of the fleet represents healthy, non-anomalous behavior. In other words, we assume that at any one time most turbines exhibit healthy behavior, an assumption that has been made before in related work (Chesterman et al., 2021; Hendrickx et al., 2020a; Beretta et al., 2020). If the fleet size is so insufficient that this assumption does not hold, this filtering method will not be effective. However, this assumption is supported by the fact that even in offshore wind farms, which experience significantly higher failure rates than onshore installations (Carroll et al., 2016), fleet availability for observed offshore farms remains above 80% as noted by Pfaffel et al. (2017), ensuring that sufficient turbines remain operational to maintain reliable central tendency measurements. When working with smaller fleets where there is a risk that this assumption does not hold at all times, it could still be possible to employ this method together with rule-based safeguards similar to what was done by Chesterman et al. (2023).

This method also implicitly assumes that the anomaly scores already account for operational particularities unique to different turbines, like downstream-positioned turbines influenced by the wake of other turbines. These environmental and operational variations can have a significant impact on turbine behavior which has driven innovative new techniques to take these factors into account. (Lin et al., 2022)

If these assumptions hold, it means that anomaly scores close to the median have a higher chance of being false positives, and scores far from the median have a higher chance of being true positives. The method adds weight to persistent deviations by accounting for temporal patterns, even if initially subtle. Incidentally, this also corrects for some false positives caused by fleet anomalies, i.e., anomalies caused by temporary situations affecting most of the fleet, such as extreme weather events or seasonal variations in the case of wind turbines.

# 3.3.1 Distance to Fleet Median





First, we must define the distance  $\delta$  between the anomaly score Anom( $\mathbf{F}_t^s$ ) for the signal s of a single turbine T and the fleet median of the anomaly scores for that signal Mdn(Anom( $\mathbf{F}_t^s$ )) at a single timestamp t. The most straightforward distance metric 300 is the difference.

$$\mathbf{F}_{t}^{s} = (\{\mathbf{T}_{1}^{s}(t), \mathbf{T}_{2}^{s}(t), \dots, \mathbf{T}_{n}^{s}(t)\}) \tag{2}$$

$$\delta_t = |\mathsf{Anom}(\mathsf{T}_t^s) - \mathsf{Mdn}(\mathsf{Anom}(\mathsf{F}_t^s))| \tag{3}$$

We first define our fleet values for a specific signal and timestep in (2), after which we subtract the median of the anomaly scores of the fleet from our turbine score (3). This is easy to calculate but intuitively flawed in several ways. For example, it does not account for potential time lag across the fleet or localized outliers.

# 3.3.2 Windowing

We can mitigate this issue by using a rolling window and aggregating the scores somehow, e.g., by defining RollAvgAnom(x,t,w) as the rolling average of the anomaly scores at time t over signal s with window size w (4), and then again taking the difference (5).

$$\operatorname{RollAvgAnom}(s,t,w) = \frac{1}{w} \sum_{i=t-(w/2)}^{t+(w/2)} \operatorname{Anom}(s(i))$$
 (4)

$$\delta_t = |\text{RollAvgAnom}(\mathbf{T}^s, t, w) - \text{Mdn}(\text{RollAvgAnom}(\mathbf{F}^s, t, w))|$$
(5)

This is a better representation to use, but it is still limited and very dependent on the chosen w. A large window might smooth a localized but severe anomaly too much, while a small window might not capture a string of severe but slightly spread-out anomalies.

# 315 3.3.3 Multidimensional distances

**Figure 4.** To construct the tuples the anomaly scores are first split by level, and then for each level, subtuples are constructed for each timestamp through multiple different-sized sliding windows. These subtuples are then joined together to form the final tuple.

To account for this, we shift our metric from calculating the difference between singletons to the difference between n-tuples, where each element is the result of RollAvgAnom (RAA) at the same t with a different interval w. This gives us a solid way to incorporate time in our comparison. If we then first define said tuple a (6), we can calculate the difference again.

$$\mathbf{a}_t^T = (\mathsf{RAA}(\mathsf{T}_1^s, t, 1D), \mathsf{RAA}(\mathsf{T}_2^s, t, 2D), \dots \mathsf{RAA}(\mathsf{T}_n^s, t, nD)) \tag{6}$$

$$\delta_t = \left| \mathbf{a}_t^T - \mathrm{Mdn}(\mathbf{a}_t^F) \right| \tag{7}$$

We can then also adjust it based on the specific format of our anomaly scores. In our case, at any timestamp, the anomaly score may be one of these values: [-3, -2, -1, 0, 1, 2, 3]. With the current n-tuple construction, we lose some nuance by throwing each "level" together, and since we also have negative anomalies, we risk averaging out important outliers. If, instead, we separate them, we can construct an expanded tuple of size  $w \cdot l$ , with w the number of windows and l the number of levels we want to include:

$$\mathbf{a} *_t^T = \mathbf{a}_t^{T[-3]} \oplus \mathbf{a}_t^{T[-2]} \oplus \dots \mathbf{a}_t^{T[m]}$$

$$\tag{8}$$

$$\delta_t = \left| \mathbf{a} *_t^T - \mathsf{Mdn}(\mathbf{a} *_t^F) \right| \tag{9}$$

Not only does this allow us to calculate the difference for separate levels, but it also enables us to exclude irrelevant scores. For example, we may only be interested in positive anomalies; if we construct our tuples this way, it is trivial not to include those scores when calculating the difference. Also, note that the manner in which we construct the tuple creates implicit assumptions about the relative difference between anomaly levels; we chose to sum the anomalies to aggregate them. This inherently equates 1 level 3 anomaly to 3 level 1 anomalies. Depending on the method of outlier detection used to produce the anomalies, it may be better to weigh the levels differently and/or use different aggregation methods.

#### 3.3.4 Distance Calculation




Now that we have constructed our tuples, we must calculate the distance between a signal and the signal fleet median. Note that the signal fleet median is calculated by constructing the tuples for each timestamp for every wind turbine and then taking the median component-wise. We used the component-wise median as an uncomplicated starting point and found it performed very well, leading us to keep using it instead of the computationally more expensive geometric median. However, using the geometric median might be a potential improvement. There are several methods to calculate distances between two tuples. We examined four specifically: Euclidean distance as a baseline, maximum distance, Manhattan distance, and finally, Mahalanobis distance as a more sophisticated metric.

We chose Euclidean distance as it is the most commonly used distance metric and is easy to calculate. It measures the linear distance between two points in a multidimensional space. However, since it assumes that all dimensions are equally important, we expected a lackluster performance in this case since we suspect that different dimensions have different levels of importance.

The Maximum distance takes the maximum difference between the corresponding coordinates of the two tuples. We chose this to determine the impact of the most significant difference in any dimension on the calculated distance. Manhattan distance is included to account for the high dimensionality and potential difference in importance of each dimension; as such, we suspected it may be more suitable in this case.

Finally, we used the Mahalanobis distance, a more sophisticated method that considers the correlations between dimensions. In contrast to the other distance calculations, the Mahalanobis distance determines the distance between a point and a distribution. So, instead of calculating the fleet median of the signal tuples, we use every tuple in the fleet at a single timestamp as the distribution. Though it is more costly to calculate, we chose this method because it is unitless and scale-invariant, which means it considers the variance and covariance of the dataset. We hoped this would provide a more accurate distance measure in the presence of correlated variables.

# 3.3.5 Thresholds





Now that we have several distance measures that we can use to compare a single wind turbine to the central tendency of the whole fleet, we encounter the problem of determining precisely what these distances mean. As our method aims to be a simple, cost-effective post-processing technique, we kept things simple and examined three different ways to set a threshold to detect the outliers that would indicate true positives.

So, after we have calculated a distance  $\delta$  for every timestamp t, we need to choose a threshold  $\tau$  to categorize anomalies at t so that an anomaly score  $a_t$ , for which  $\delta_t > \tau_t$ , is deemed significant. The most straightforward way to obtain a value for  $\tau$  is by calculating the  $n^{\text{th}}$ -percentile for a healthy subset of all distances. Such a subset can be obtained relatively easily by using maintenance reports to discard all wind turbines with known failures and ranking the remainder by the amount and severity of detected anomalies. Then, we can use the calculated percentile as a constant threshold; in our results, we calculated and used this for the  $95^th$ -percentile (const95%).

Another method that can be used to set a threshold is calculating the  $n^{th}$ -percentile  $(P_n)$  for every t across all turbines T so that:

$$\tau_t = P_n(\{\delta_t^{T1}, \delta_t^{T2}, \ldots\}) \tag{10}$$

This produces a threshold that varies with every timestamp and can be straightforwardly tuned by changing the desired percentile based on how conservative you want to be (Leys et al., 2013). In our results, we calculated and used this for the  $95^{t}h$ -percentile (Var95%). Similarly, we also used a modified Median Absolute Deviation (MAD) to set another variable threshold. So that: for every t across all turbines T and median Mdn, with

$$MAD = \operatorname{Mdn}(|X_i - \operatorname{Mdn}(X_t)|) \tag{11}$$

being the formula for the MAD and  $\delta_t$  already being a measure of the absolute distance to the median, we defined the threshold  $\tau$  as shown in (12), substituting  $|X_i - \text{Mdn}(X_t)|$  and approximating the std with b = 1.4826 (Rousseeuw and Croux, 1993).

$$\tau_t = n \cdot (b \cdot \text{Mdn}(\{\delta_t^{T1}, \delta_t^{T2}, \dots\})) \tag{12}$$

In our results, we calculated and used this for n = 3 (MAD $\times$ 3).

# 380 **3.3.6** Validation

It is difficult to determine whether an outlier is significant, so to validate our method and show that it mainly filters out non-significant scores, we have taken two approaches. Firstly, we employed a heuristic to choose Healthy and Unhealthy zones in our dataset, which is similar to the method employed by Chesterman et al. (2023); secondly, we empirically analyzed several failure cases and looked at how well our method predicted them based on the available anomaly scores.

# 385 Healthy and unhealthy zones

To show that our method does not filter out a significant amount of failure-predicting anomalies, we attempt to show that it is statistically grounded. To do this, we are forced to make some assumptions. First, we assume that we can find healthy and unhealthy periods in our data. Data is healthy if it does not exhibit abnormal behavior caused by a damaged component and unhealthy if the reverse is true. Second, we assume that a certain period preceding a known failure is unhealthy. Then we used these assumptions to define healthy and unhealthy zones of near equal size, compared the observed anomalies, and calculated the distances to the fleet median in each zone.

Before marking the healthy zones, we also introduced a buffer period after the end of a failure, as signals tend to be quite erratic during that time. Since we examined multiple relatively recent failures, we could not choose the year after failure as a Healthy zone as done in Chesterman et al. (2021). Instead, we chose a period preceding an unhealthy zone as the healthy zone, adding a small buffer zone in between. This healthy zone is equal in size to the unhealthy zone and does not override another buffer or unhealthy zone.

We expected and confirmed that we would find more, and more severe, anomalies in unhealthy zones compared to healthy ones. Then, we examined the calculated distances in those zones and established the same relation. Showing that we chose zones correctly and that our distance calculation accurately reflects the detected anomalies.

# 400 Empirically



However, comparing the distribution of anomalies across these zones before and after filtering provides limited insight into how much of a signal's predictive power is preserved. Consequently, to validate our method further, we also examined the impact of filtering on several signals known to predict a generator or gearbox failure accurately. Assuming that, if the filtered

anomaly scores demonstrate a significant decrease in predictive accuracy compared to the unfiltered scores, the method would be critically flawed.

# 4 Results

In this section, we present our results, describe the data we used, and explain how we generated the anomaly scores. We discuss the performance of the undercomplete autoencoder before moving on to our proposed anomaly filtering method. We show our validation plots and then compare and discuss the various submethod combinations.

# 410 4.1 Turbine Data

All results were obtained using 10-minute SCADA (Supervisory Control and Data Acquisition) data, an industrial control system widely used to manage and monitor critical infrastructure, including offshore wind turbines. This data was collected over approximately 4 years from a large (>20 turbines) wind farm in the Dutch-Belgian offshore zone. All turbines are rated for more than 8MW. We used a subset of the available signals, focusing mainly on the temperature signals of the generator (GEN) and the gearbox (GBX), along with more standard signals such as windspeed, power, rotor speed, etc. We used partially human-written maintenance logs and reports documenting eight gearbox failures and five generator failures. The failure types for all generator failures are short-circuits due to mechanical causes, and the gearbox failures are caused by the bearing, more information cannot be disclosed due to confidentiality agreements. These records informed our selection of training data and helped to validate our results.

# 420 4.2 Framework



In this section, we present the deployment results of the scalable pipeline framework, demonstrating its performance across two offshore wind farms and evaluating its operational characteristics in both initial deployment and continuous operation modes. We examine the framework's scalability through parallel fleet deployment and analyze the available runtime performance metrics. The scalable pipeline framework was successfully deployed across two distinct offshore wind farms in the Dutch-Belgian offshore zone, each comprising more than 20 turbines with approximately four years of operational data.

A typical deployment begins with local prototype development and testing, progressing through automated containerization to full cloud deployment. We adapted an existing implementation of NBM anomaly detection to make use of the pipeline framework. Once the prototype code achieved satisfactory performance on local infrastructure, we generated the necessary configuration files and directed the framework to initiate the deployment sequence. This started a local process that leveraged Docker containerization technology to package pipeline modules as images, which were then uploaded to a remote Docker repository. This approach ensured consistent deployment environments and simplified the scaling process across different wind farm configurations. After uploading the containerized modules, the framework interfaced with a remote Kubernetes cluster to initiate the overwatch pod, which assumed responsibility for the further execution of the pipeline. It created the

required storage directories on the Hadoop HDFS infrastructure and orchestrated module execution according to the specified configuration.





The framework's scalability was validated through parallel deployment across two offshore wind farms, rather than intrafarm scaling (parallelization within a single deployment), due to the pre-existing pipeline implementation constraints. While the original NBM implementation was not specifically designed to leverage the framework's full distributed processing capabilities, the parallel deployment provided valuable insights into the system's horizontal scaling potential. This parallel scaling approach demonstrated that the framework can effectively manage multiple independent wind farm instances simultaneously, with each deployment maintaining isolated data processing pipelines while sharing the underlying cloud infrastructure. Note also that the farms selected for deployment are sufficiently similar in characteristics to allow for a meaningful comparative analysis, making the parallel deployment results representative of what could be achieved in intra-farm scaling scenarios. As can be seen in Table 1, deploying the framework for training for both farms in parallel took approximately 26.5 hours. If run in series, the required time would be around 46 hours roughly. Whether these timesavings can be achieved or exceeded through intrafarm scaling depends on the chosen anomaly detection model and implementation. In this specific case, it would be possible to make significant gains by training each sub-model from the ensemble in parallel.

| Run        | Data cleaner |        | Preprocessor |        | Anomaly detection |        | Failure diagnosis |        | Total<br>wall- | Assigned re- |
|------------|--------------|--------|--------------|--------|-------------------|--------|-------------------|--------|----------------|--------------|
|            |              |        |              |        |                   |        |                   |        | time           | sources      |
|            | WT           | core-h | WT           | core-h | WT                | core-h | WT                | core-h |                |              |
| Train A    | 2 min        | 2.1    | 3 min        | 3.1    | 19:25:00          | 1203.8 | 11 sec            | 0.2    | 19:31:25       | 62 cores     |
| Train B    | 8 min        | 9.6    | 19 min       | 22.8   | 25:59:36          | 1870.8 | 11 sec            | 0.2    | 26:28:29       | 72 cores     |
| Cont. A*   | 1 min        | 1      | 6 min        | 6.2    | 14 min            | 14.5   | 2 min             | 2.1    | 00:43:19       | 62 cores     |
| Autoencode | er** –       | _      | _            | _      | 00:12:04          | 4      | _                 | _      | _              | 20 cores     |

Table 1. Computational cost breakdown for each pipeline module showing wall-time (WT) and core-hours (core-h). Core-hours are rounded up to one decimal place. Processor types and clocks are reported in Table A1. Note that the total wall time includes cluster management overhead and that the core-h for the Data cleaner, Preprocessor, and Failure diagnosis are pessimistic estimates since these modules do not always use all the available cores. RAM was never a bottleneck and is not shown. Train A and B are deployments for farms A and B, where the shallow ML models are trained before generating the error scores. \*Continuous mode, one iteration where the pipeline pulls and processes new data generated during one day and updates the existing results. \*\*Added the autoencoder model from Sect. 3.1.3 for comparison; Metrics shown are for 200 epochs, a normal training run goes for 250 epochs with early stopping; This module was not run using the framework, and without parameter tuning, with parameter tuning a training run takes on average less than 5 hours.

Table 1 shows the time and core hours for three deployments of the pipeline framework using an ensemble of statistical models for anomaly detection. Two deployments were started simultaneously for farms A and B, Farm A is the same farm described in Sect. 4.1 and Farm B is another farm in the Dutch-Belgian offshore zone with approximately double the number

of turbines. The third deployment is for farm A in continuous mode, where the deployed pipeline will ingest new data every day and use the pre-trained models generated during the training deployment to calculate the results for the new data and update the preexisting results. This is the reason for the drastically lower core-h for the anomaly detection. Similarly, the preprocessor and failure diagnosis for the continuous deployment take longer than for the training deployment since they need to update the original results, causing more overhead due to file accessing.

# 4.3 Autoencoder






The autoencoders can accurately model the normal behavior of the wind turbines. To this end, the Mean Absolute Reconstruction Error (MARE) (which is the average of the absolute values of the reconstruction errors) of the different relevant temperature signals is calculated on healthy data that has not been used for hyperparameter tuning. The generator model achieves an average generator signal MARE of 0.40 °C, with a maximum of 0.78 °C. For the gearbox model, the average signal MARE is 0.39 °C, with a maximum of 0.49 °C.

Figures 5, 6, and 7, show the evolution of the reconstruction error for three different wind turbines just before the occurrence of a major generator failure. The reconstruction error in the figures is calculated as the difference between the predicted and the observed temperatures of the signals. If the observed temperature is higher than the temperature predicted by the NBM, then the reconstruction error is positive. If the opposite is the case, the reconstruction error is negative. By defining the reconstruction error like this, it becomes clearer when the temperatures of the wind turbine components are above expected and when below. The former case is interesting because it can be indicative for component damage, the latter can point to model imperfections, which can then be analyzed further.

In Figures 5 and 7 it is clear that already several months before the failure the temperatures of the generator phases behave out of the ordinary. The observed phase temperatures are substantially higher than what is predicted. For the wind turbine in Fig. 6 the evolution is somewhat less clear. Just before the failure, there is a peak in the reconstruction error. In this case, the problem is only detected a couple of days in advance.

The reconstruction error is the basis for the anomaly detection. To be suitable, it must be small during healthy periods and large just before the failures. An interesting metric for this is the UHH-ratio (UnHealthy-Healthy). It calculates the ratio of the MAREs on unhealthy and healthy data. A useful NBM has a UHH-ratio substantially larger than 1, which means that its MARE on unhealthy data is larger than that on healthy data.

For the generator model, the average of the UHH-ratios of the generator temperature signals is 1.40, with a maximum value of 1.69. For the gearbox model, the average of the UHH-ratios of the gearbox temperature signals is 1.08, with a maximum value of 1.21. The relatively low average UHH-ratio is not necessarily a problem. It is possible that of the 5 gearbox temperature signals only one signal really captures the problem. For this reason, the maximum UHH-ratios are more relevant.

#### 4.4 Farm-wide Anomaly Filtering

In this section, we present our analysis of the filtering technique we developed. We validate this method by showing how it primarily removes anomalies that are unrelated to an upcoming failure while keeping the significant anomalies. Furthermore, we

**Figure 5.** Figure shows the reconstruction error for a generator phase temperature signals of a wind turbine that experienced some time after month 9 (calculated from the start of the figure) a generator failure.

**Figure 6.** Figure shows the reconstruction error for a generator phase temperature signals of a wind turbine that experienced some time after month 4 (calculated from the start of the figure) a generator failure.

empirically evaluate the results of applying different distance calculation methods and thresholds. We examine its performance on a specific generator failure case and, more generally, on a whole fleet. Note that the Anomaly scores used in this section were generated by the autoencoder-based NBM (Sect. 3.1.3) and are the same as those evaluated in Sect. 4.3. Note that the fleet size was large enough with few enough failures that the fleet median was not affected by turbine downtime. Furthermore, the autoencoder-based NBM mostly accounts for significant turbine-specific particularities, such as position-based wake influence. Since it uses signals like Rotor speed, Active power, and Wind speed as predictors. This means the results already account for the significant position-based quirks of the turbines.



**Figure 7.** Figure shows the reconstruction error for a generator phase temperature signals of a wind turbine (not the same wind turbine as in Fig. 5) that experienced some time after month 9 (calculated from the start of the figure) a generator failure.

# 4.4.1 Validation

# Zones




We looked at data from all turbines with a known generator or gearbox failure to validate our method. We chose zones as explained in Sect. 3.3.6. Concretely, we marked the 210 days preceding a failure as unhealthy and the 210 days before that unhealthy period as healthy, with a 60-day buffer in between. Also note that we maintained a 60-day buffer after the end of a failure, as signals tend to be very unstable during that time. We also defined a Healthy-turbine zone by choosing the three healthiest turbines. This was done by ranking all turbines by the number of level three anomalies and picking the three lowest.

Looking at the sample data for all signals of turbines with known generator or gearbox failures in Fig. 8a, we see that, as we expected, the amount of anomalies is significantly higher in the Unhealthy zone compared to the Healthy zone, suggesting our Healthy and Unhealthy zones are well chosen.

We also note that the anomaly count in the Healthy zone is slightly higher than in the Healthy-turbine zone, representing the presence of non-relevant (for predicting failure of our chosen components) anomalies. Though we chose the zones manually in this case, it might also be possible to automatically select the Unhealthy and Healthy zones based on comparing the Healthy and Healthy-turbine zones. This would allow us to calculate a threshold (see Sect. 3.3.5) instead of manually choosing it.

In Fig. 8b, we examine the distribution of anomalies per zone. Again, as we expected, the Unhealthy zone has a much larger proportion of higher-level anomalies than the Healthy zone. Placing the median much higher. The distribution of the Healthy zone and the Healthy-turbines are nearly identical, though the average in the Healthy zone is slightly higher.

If we look at the distribution of calculated distances in Fig. 10, we observe a similar division between zones for each distance metric, showing that our tuple construction and distance calculation methods accurately reflect the original anomaly scores.

If we then compare Figs. 8 with Figs. 9, which shows the same plots, but for the filtered anomaly scores, we can see that the Unhealthy zone is far less affected compared to the Healthy and Healthy-turbine zone. This aligns with our expectations, as

**Figure 8.** (a) Bar plot showing the count of absolute anomaly levels from equally long random samples taken from Healthy, Unhealthy and Healthy-turbine zones. (b) Violin plot showing the distribution of the absolute anomaly scores in Healthy, Unhealthy and Healthy-turbine zones for all scores from all turbines with known generator or gearbox failures.

**Figure 9.** For these plots, the anomaly scores have been filtered using Manhattan distance, with the tuples constructed using window sizes: (1D, 5D, 10D, 20D) and all anomaly levels: (-3,...,3). (a) Bar plot showing the count of absolute adjusted anomaly levels from equally long random samples taken from Healthy, Unhealthy and Healthy-turbine zones. (b) Violin plot showing the distribution of the absolute adjusted anomaly scores in Healthy, Unhealthy and Healthy-turbine zones for all scores from all turbines with known generator or gearbox failures.

we would indeed expect the healthy zone to contain few relevant anomalies. Furthermore, since we anticipate that the Healthy-turbine zone would have virtually no relevant anomalies, given that this zone was sampled from turbines with no failures, the notably low anomaly count we can see is fully consistent with this expectation. A better breakdown of the removed anomaly scores can be seen in Fig. 15.


**Figure 10.** Violin/box plots showing the distance to the fleet median in Healthy and Unhealthy zones for all distance calculation methods, tuples were constructed using window sizes: (1D, 5D, 10D, 20D) and all anomaly levels: (-3,...,3).

**Figure 11.** Violin/box plots showing the distance to the fleet median in Healthy and Unhealthy zones using Manhattan distance for specific signals, tuple was constructed using windowsizes: (1D, 5D, 10D, 20D) and all anomaly levels: (-3,...,3).

Figures 8 and 10 take into account the anomaly scores of all signals. However, the distribution of individual signals can differ greatly, as can be seen in Fig. 11. This is caused by the fact that not all signals predict an oncoming failure and that failures may be characterized by different combinations of anomaly severities, as discussed in Sect. 3.3. This emphasizes the importance of having a large enough fleet so that the quirks are normalized and the median becomes reliable.




**Figure 12.** Original anomaly scores and filtered anomaly scores aggregated with a rolling average per day. The filtering was done with tuples constructed out of all levels (-3 to 3), using Manhattan distance and the var95% threshold. The striped black line is a generator failure start.

Figure 12 shows the effect of filtering the original anomalies of a composite signal. The original scores show a noticeable increase as the failure approaches. The filtered anomalies still clearly predict and emphasize the coming failure, while much of the noise has been removed. Fleet-wide, we can identify 30 occasions where a (singular) signal can be said to accurately and significantly (however minorly) predict an oncoming failure. Of those occasions, there are at least 22 where the filtered anomaly scores maintain the same predictive accuracy as the original scores. Note that the eight signals, for which this is not true, show only marginal failure prediction in the unfiltered anomaly scores. Note that filtering out so many anomalies leads to a compression of the x-axis, with the remaining (averaged by day) values being, on average, lower than the original anomaly scores.

# 4.4.2 Submethod Comparisons

#### Tuple Construction

How one constructs the tuples affects the calculated distances. Some knowledge of the meaning of the different anomaly levels is required. If you are only interested in failures that, you are certain, are mainly predicted by positive anomalies, you may want to ignore all negative anomalies. However, note that an increase in the arity of the tuples results in a reduced sensitivity for the overall distance towards each individual component of the tuples, as the contribution of each individual term becomes less significant relative to the calculated distance. As such, if the predictive power of a specific signal mainly relies on a single component, i.e., the 5D window of level 3 anomalies. Increasing the arity will decrease the impact of that component and, thus, its predictive power. This can be counteracted by adding weights to the tuple elements. However, if you do not have accurate

|                  |         | Variable 95% threshold |         | Constant 95 | % threshold | Variable MAD×3 threshold |         |
|------------------|---------|------------------------|---------|-------------|-------------|--------------------------|---------|
|                  | Tuple   | Unhealthy              | Healthy | Unhealthy   | Healthy     | Unhealthy                | Healthy |
| Orig. scores     |         | 106452                 | 16980   | 106452      | 16980       | 106452                   | 16980   |
| Euclidean        | -3223   | 85036.0                | 4751.0  | 97368.0     | 10998.0     | 92577.0                  | 7556.0  |
| Euchdean         |         | -20.12%                | -72.02% | -8.53%      | -35.23%     | -13.03%                  | -55.50% |
|                  | -321123 | 85746.0                | 5557.0  | 98085.0     | 10904.0     | 91336.0                  | 6917.0  |
|                  |         | -19.45%                | -67.27% | -7.86%      | -35.78%     | -14.20%                  | -59.26% |
| Manhattan        | -3223   | 85756.0                | 4669.0  | 98231.0     | 11575.0     | 94108.0                  | 8298.0  |
| Maiinattan       |         | -19.44%                | -72.50% | -7.72%      | -31.83%     | -11.60%                  | -51.13% |
|                  | -321123 | 86692.0                | 5557.0  | 99164.0     | 11589.0     | 92111.0                  | 7555.0  |
|                  |         | -18.56%                | -67.27% | -6.85%      | -31.75%     | -13.47%                  | -55.51% |
| M :              | -3223   | 84845.0                | 4470.0  | 97125.0     | 11026.0     | 92210.0                  | 7495.0  |
| Maximum _        |         | -20.30%                | -73.67% | -8.76%      | -35.06%     | -13.38%                  | -55.86% |
|                  | -321123 | 85142.0                | 5389.0  | 97209.0     | 10364.0     | 90877.0                  | 6894.0  |
|                  |         | -20.02%                | -68.26% | -8.68%      | -38.96%     | -14.63%                  | -59.40% |
| Mahalanobis      | -3223   | 83517.0                | 6304.0  | 81565.0     | 7196.0      | 85230.0                  | 7818.0  |
| Mahalanobis<br>- |         | -21.54%                | -62.87% | -23.38%     | -57.62%     | -19.94%                  | -53.96% |
|                  | -321123 | 84337.0                | 5785.0  | 88828.0     | 8550.0      | 90917.0                  | 7739.0  |
|                  |         | -20.77%                | -65.93% | -16.56%     | -49.65%     | -14.59%                  | -54.42% |

**Table 2.** Comparison of the effect of tuple dimension size for all distance methods and thresholds. Shows absolute total value  $(\sum |Anomaly\ score|)$  and the percentage difference compared to the original scores.

information about the relative importance of each element, experimentation may be required to determine adequate weights. We did not apply weights for the results shown here, as this method is intended to be straightforward and robust. We evaluated its effectiveness with minimal adjustments, only tuning the window sizes and included anomaly levels to a limited extent.



In Table 2, we compare the adjusted anomaly scores generated using tuples with and without including level 1 (1 and -1) anomalies. The thresholds used are explained in Sect. 3.3.5. We see that including the less severe anomalies caused fewer anomalies to be filtered out. On average, constructing the tuples with all available anomaly levels caused  $1.748\% \pm 2.025\%$  fewer anomalies to be removed in the unhealthy zone and  $3.591\% \pm 2.216\%$  in the healthy zone.

Examining the differences empirically, we found that the observed results could vary noticeably in a few cases. Excluding level 1 anomalies notably increased the visibility of failure-predicting scores for some turbines. This is likely due to the decreased arity and the fact that these specific failures have many high-level (2 and 3) anomalies. However, as expected, other

failure predictions more characterized by a large number of low-level anomalies saw a decrease in visibility when the low levels were left out of the used tuples.

# 550 Contrast Distances


**Figure 13.** Filtered anomaly scores aggregated with a rolling average per day. The filtering was done with tuples constructed from all levels (-3 to 3), using the var95% threshold. The striped black line is the generator failure start.

Figure 13 illustrates the effect of different methods used to calculate the distance to the median for our turbine. The visible differences are minimal in this case, though Mahalanobis does filter noticeably more aggressively. Looking more generally, we find that, as can be expected, distances in the unhealthy zone are drastically higher than in the healthy zone for Euclidean, Manhattan, Maximum, and Mahalanobis (Fig. 10). Using Euclidean and Manhattan generally produces similar results, while Maximum behaves as if it has a much lower arity and is occasionally less sensitive. Mahalanobis is the odd one out. Though still generally similar, it is sometimes more and sometimes less sensitive than the others. We suspect this is due to its disproportionate sensitivity to more severe anomaly levels. Weighting the anomaly scores as discussed earlier may make it more stable.

**Figure 14.** Filtered anomaly scores aggregated with a rolling average per day. The filtering was done with tuples constructed from all levels (-3 to 3), using the Manhattan distance. The striped black line is the generator failure start.

#### Contrast Thresholds



Figure 14 illustrates the effect of different threshold choices as explained in Sect. 3.3.5: var95% is the strictest, filtering out most anomalies, and const95% is the loosest overall. However, const95% seems to fluctuate, sometimes stringent, other times not. This is likely a result of being a constant threshold, which sometimes results in the removal of significant predictive data, as seen in the figure; const95% filters out too much, removing all failure-predicting anomalies. MAD×3 removes fewer anomalies overall than var95% but overfocuses on the failure-predicting anomalies.

These observations generally hold when we examine all the results, as shown in Fig. 15; refer to Table C1 for a full breakdown. Of the three thresholds examined, we can say that the simple Var95% avoids the inconsistency of Const95% while removing more anomaly scores overall compared to MAD×3. But removing less relevant scores, thus maintaining a higher prediction accuracy. It is unexpected that MAD×3, which is nominally a more severe threshold, filters less anomaly scores overall. Examining C1 shows us that the lower scores (-2,..., +2) are the most affected, while Var95% scarcely filters out more (-3,+3) scores compared to MAD×3. This less strict thresholding of MAD×3 is most likely due to our modified MAD×3 no longer corresponding to 1·std, and consequently being less severe than the 95<sup>th</sup> percentile.

**Figure 15.** Comparison of distance methods and thresholds. Shows the percentage difference in number of anomaly scores compared to the number of original scores in the healthy and unhealthy zones. Considers all nonzero scores of healthy and unhealthy zones (as defined in Sect. 3.3.6) of all turbines with known GEN or GBX failures.

# 4.5 Discussion



Our analysis of the anomaly filtering technique resulted in several insights regarding methodological choices, which have implications for the practical application of our method.

From our comparisons, we can conclude that the choice of distance metric and thresholding strategy plays a crucial role in balancing noise reduction with the preservation of predictive signals. Our aim was to minimize any fine-tuning since this filtering method is designed to be straightforward and robust, so we expressly examined the base capabilities of each approach. This resulted in the Manhattan method emerging as the most effective distance calculation, providing robust performance in filtering out non-relevant anomalies. In contrast, while the Mahalanobis method shows potential, the results suggest it requires more fine-tuning to achieve consistent sensitivity, especially given its variable response to severe anomaly levels.

Regarding thresholding, the constant threshold approach is outperformed by dynamic thresholds. Among the dynamic strategies, the 95<sup>th</sup> percentile approach (var95%) is particularly noteworthy. Although it removes a slightly higher number of anomalies overall compared to the MAD×3 method, even in the unhealthy zones, the increased removal mainly affects lower-severity

anomalies. Coupled with the var95% threshold being more sensitive to the underlying distribution at each timestep, meaning that it adapts better to fluctuations in the data (e.g., a temporary abundance of outliers), it results in var95% being better at preserving the anomalies most critical for failure prediction while still filtering out the largest amount of anomalies.

Our anomaly filtering method offers several practical benefits that contribute to more efficient and effective operations. It is relatively computationally lightweight, scaling approximately log-linearly with the number of turbines, and multiplicatively with the dimensionality, i.e., the number of time windows and the number of anomaly levels considered. By reducing the volume of raw anomaly scores by up to 65%, the technique significantly diminishes noise (also including non-relevant anomalies in that term) and enhances data visualization, making it easier for operators to identify events of interest at a glance. This noise reduction also simplifies the implementation of automatic alarming. With only the most significant anomalies remaining, the threshold-setting process becomes more straightforward. This ability to obtain a rapid initial overview of the system's state with considerably less effort is particularly advantageous in large-scale operations where one needs to consider extensive amounts of data for analysis. This also makes detecting and reacting to significant anomalous events easier and faster.

Additionally, the filtering method's reliance on the fleet median naturally filters out farm-wide anomalies like seasonal variation and extreme weather. Furthermore, because the method only removes anomalies, the risk of false positives is maintained at the same level as the original anomaly scores. Finally, while there exists a potential loss of true positives, we found that it is mitigated by the fact that multiple signals predict any GEN or GBX failure. This provides redundancy, as the chance that filtering causes all these signals to lose their relevant anomalies is minimal. In our experiments, after filtering, all failures were still predicted by at least one signal or two for the GBX and GEN failures, respectively.

# 5 Conclusions







In this study, we examined solutions to address the need for scalable and effective anomaly detection in offshore wind turbines. Such solutions enhance operational reliability and reduce maintenance costs, which is particularly important given the increasing demand for renewable energy and offshore wind farms. We deployed a scalable cloud-based pipeline fit for packaging an implementation of the NBM framework to perform failure detection across multiple offshore farms. We further introduced and evaluated an autoencoder-based NBM implementation, as well as a statistical filtering technique to systematically remove non-relevant anomalies. Improving efficiency when analyzing the output of anomaly detection.

Our pipeline enables scalable and rapid analysis across multiple wind farms while supporting diverse NBM implementations and configurations. The modularity of the architecture allows individual components to be easily interchanged or upgraded without disrupting the entire system. This flexibility enables experimentation with different models and parameterizations tailored to specific operational conditions. Furthermore, it allows for automatic hyperparameter optimization tailored to individual farms, mitigating the impact of site-specific variations on anomaly detection performance. Additionally, the cloud-based implementation supports dynamic scaling, allocating computational resources as needed during training and inference. This ensures efficient processing even as the volume of incoming data increases. The pipeline further facilitates seamless deploy-

ment and maintenance, with automatic packaging of module code and integration with container orchestration technologies such as Kubernetes.

We also implemented an anomaly detector according to the NBM framework. Specifically, we trained an undercomplete autoencoder, and though it should be able to learn with unlabeled data, we posited that training it exclusively on healthy data might increase its performance. To do this, we curated the available data based on a failure list provided by the turbine generator and a forced shutdown list we extracted from the status logs. Our generator model demonstrated a UHH-ratio of up to 1.69, while the gearbox model exhibited a UHH-ratio of up to 1.21.






A key innovation of this study is our time-aware anomaly filtering method, which refines anomaly scores using the fleet median as a reference. This technique reduces the volume of raw anomaly scores by up to 65% while preserving nearly identical predictive accuracy, facilitating straightforward analysis and more efficient alarming. Properly configured, this approach retains significantly predictive anomalies while minimizing noise and anomalies unrelated to the targeted failures. It also enhances the failure predictions for some anomaly detection methods through mitigating the impact of fleet-wide anomalies. To achieve this, we aggregated the original anomaly scores using sliding windows of different sizes. We then calculated the distance between the resulting tuples and the fleet median, creating a time-aware measure of the difference between a turbine and the fleet for any specific signal, enabling more precise anomaly filtering.

Our analysis of distance metrics and thresholding strategies identified Manhattan distance as the most robust measure. However, Mahalanobis distance, which accounts for variance and covariance among dimensions, exhibited superior performance in some cases, suggesting potentially better performance with further tuning. The other metrics, Euclidean and Maximum distance, tended to be less sensitive to low-level, but persistent and numerous, anomalies. We also showed that threshold selection plays a crucial role, finding that the Var95% threshold provided the best balance between filtering stringency and predictive accuracy, outperforming the alternative approaches Const95% and MAD×3. This thresholding approach effectively filters out non-relevant data while preserving significant anomalies, leading to more reliable maintenance alerts. And since our filtering method is purely subtractive, it ensures that the probability of false positives remains equal to or lower than the original anomaly scores.

In summary, we presented a scalable pipeline that enables us to swiftly and simply scale failure detection across offshore wind farms. We also presented an undercomplete autoencoder-based NBM implementation that performs temperature-based anomaly detection. Furthermore, we proposed and evaluated an anomaly filtering technique based on comparing turbine signals to the fleet median and found that it significantly reduces noise while also retaining critical predictive anomalies. These advancements lead to improved predictive maintenance, reduced operational costs, and contribute to the broader goals of sustainable and cost-effective renewable energy production. Future work could focus on refining these methods, further automating the pipeline, enhancing the deep autoencoder method, exploring additional metrics, and optimizing filtering thresholds further to enhance the robustness and precision of the filtering process.

**Appendix A: Cluster Resources** 

| Node types | CPU type                 | CPU Clock (GHz) |  |  |
|------------|--------------------------|-----------------|--|--|
| Node 1     | Intel® Xeon® Silver 4214 | 2.20            |  |  |
| Node 2     | Intel® Xeon® Gold 6348   | 2.60            |  |  |
| Node 3     | Intel® Xeon® Gold 6130   | 2.10            |  |  |
| Node 4     | Intel® Xeon® Silver 4214 | 2.20            |  |  |
| Node 5     | Intel® Xeon® Gold 6430   | 2.10            |  |  |

**Table A1.** Processor types and clock speeds of the Kubernetes cluster used to run the pipeline framework. The autoencoder was run using the same processor type as Node 5.

# Appendix B: Distance-Threshold Comparison Table

|                        |        | Variable 95% threshold |                  | Constant 95       | 5% threshold      | Variable MAD×3 threshold |                  |  |
|------------------------|--------|------------------------|------------------|-------------------|-------------------|--------------------------|------------------|--|
|                        | Score  | Unhealthy              | Healthy          | Unhealthy         | Healthy           | Unhealthy                | Healthy          |  |
|                        | -3     | 10948   20.91%         | 972   8.76%      | 10948   20.91%    | 972   8.76%       | 10948   20.91%           | 972   8.76%      |  |
|                        | -2     | 4956   9.74%           | 885   7.98%      | 4956   9.74%      | 885   7.98%       | 4956   9.74%             | 885   7.98%      |  |
| 0 : : 1                | -1     | 9855   18.82%          | 3527   31.80%    | 9855   18.82%     | 3527   31.80%     | 9855   18.82%            | 3527   31.80%    |  |
| Original anomaly score | es 1   | 9940   18.89%          | 3632   32.74%    | 9940   18.89%     | 3632   32.74%     | 9940   18.89%            | 3632   32.74%    |  |
|                        | 2      | 6085   11.62%          | 1093   9.85%     | 6085   11.62%     | 1093   9.85%      | 6085   11.62%            | 1093   9.85%     |  |
|                        | 3      | 10577   20.20%         | 983   8.86%      | 10577   20.20%    | 983   8.86%       | 10577   20.20%           | 983   8.86%      |  |
|                        | total* | 106452                 | 16980            | 106452            | 16980             | 106452                   | 16980            |  |
|                        | -3     | 9967   -8.96%          | 311   -68.00%    | 10642   -2.80%    | 689   -29.12%     | 10360   -5.37%           | 327   -66.36%    |  |
|                        | -2     | 3677   -25.81%         | 195   -77.97%    | 4595   -7.28%     | 550   -37.85%     | 4031   -18.66%           | 262   -70.40%    |  |
|                        | -1     | 5454   -44.66%         | 1348   -61.78%   | 8049   -18.33%    | 2226   -36.89%    | 6805   -30.95%           | 1817   -48.48%   |  |
| Euclidean              | 1      | 5678   -42.88%         | 1191   -67.21%   | 7507   -24.48%    | 2227   -38.68%    | 6664   -32.96%           | 1713   -52.84%   |  |
|                        | 2      | 4587   -24.62%         | 297   -72.83%    | 5447   -10.48%    | 700   -35.96%     | 4913   -19.26%           | 386   -64.68%    |  |
| _                      | 3      | 9395   -11.18%         | 367   -62.67%    | 10173   -3.82%    | 628   -36.11%     | 9633   -8.93%            | 370   -62.36%    |  |
|                        | total* | 85746.0   -19.45%      | 5557.0   -67.27% | 98085.0   -7.86%  | 10904.0   -35.78% | 91336.0   -14.20%        | 6917.0   -59.26% |  |
|                        | -3     | 9993   -8.72%          | 334   -65.64%    | 10695   -2.31%    | 738   -24.07%     | 10419   -4.83%           | 376   -61.32%    |  |
|                        | -2     | 3756   -24.21%         | 209   -76.38%    | 4673   -5.71%     | 597   -32.54%     | 4113   -17.01%           | 328   -62.94%    |  |
|                        | -1     | 5546   -43.72%         | 1369   -61.19%   | 8212   -16.67%    | 2338   -33.71%    | 7003   -28.94%           | 1923   -45.48%   |  |
| Manhattan              | 1      | 5821   -41.44%         | 1146   -68.45%   | 7684   -22.70%    | 2306   -36.51%    | 6901   -30.57%           | 1815   -50.03%   |  |
|                        | 2      | 4700   -22.76%         | 322   -70.54%    | 5545   -8.87%     | 738   -32.48%     | 5016   -17.57%           | 433   -60.38%    |  |
| _                      | 3      | 9478   -10.39%         | 326   -66.84%    | 10249   -3.10%    | 687   -30.11%     | 9564   -9.58%            | 389   -60.43%    |  |
|                        | total* | 86692.0   -18.56%      | 5557.0   -67.27% | 99164.0   -6.85%  | 11589.0   -31.75% | 92111.0   -13.47%        | 7555.0   -55.51% |  |
|                        | -3     | 9960   -9.02%          | 301   -69.03%    | 10604   -3.14%    | 679   -30.14%     | 10342   -5.54%           | 320   -67.08%    |  |
|                        | -2     | 3644   -26.47%         | 170   -80.79%    | 4546   -8.27%     | 512   -42.15%     | 3987   -19.55%           | 234   -73.56%    |  |
|                        | -1     | 5426   -44.94%         | 1341   -61.98%   | 7900   -19.84%    | 2131   -39.58%    | 6702   -31.99%           | 1828   -48.17%   |  |
| Maximum<br>—           | 1      | 5557   -44.09%         | 1197   -67.04%   | 7370   -25.86%    | 2165   -40.39%    | 6547   -34.13%           | 1700   -53.19%   |  |
|                        | 2      | 4508   -25.92%         | 273   -75.02%    | 5354   -12.01%    | 653   -40.26%     | 4860   -20.13%           | 408   -62.67%    |  |
|                        | 3      | 9325   -11.84%         | 354   -63.99%    | 10109   -4.42%    | 567   -42.32%     | 9636   -8.90%            | 374   -61.95%    |  |
|                        | total* | 85142.0   -20.02%      | 5389.0   -68.26% | 97209.0   -8.68%  | 10364.0   -38.96% | 90877.0   -14.63%        | 6894.0   -59.40% |  |
| Mahalanobis<br>—       | -3     | 9989   -8.76%          | 309   -68.21%    | 10237   -6.49%    | 618   -36.42%     | 10389   -5.11%           | 320   -67.08%    |  |
|                        | -2     | 3891   -21.49%         | 241   -72.77%    | 4275   -13.74%    | 443   -49.94%     | 4136   -16.55%           | 305   -65.54%    |  |
|                        | -1     | 6043   -38.68%         | 1494   -57.64%   | 6503   -34.01%    | 1491   -57.73%    | 6887   -30.12%           | 2089   -40.77%   |  |
|                        | 1      | 5521   -44.46%         | 985   -72.88%    | 6050   -39.13%    | 1560   -57.05%    | 6544   -34.16%           | 1879   -48.27%   |  |
|                        | 2      | 4555   -25.14%         | 314   -71.27%    | 4965   -18.41%    | 493   -54.89%     | 4913   -19.26%           | 484   -55.72%    |  |
|                        | 3      | 8638   -18.33%         | 423   -56.97%    | 9028   -14.64%    | 591   -39.88%     | 9407   -11.06%           | 411   -58.19%    |  |
|                        | total* | 84337.0   -20.77%      | 5785.0   -65.93% | 88828.0   -16.56% | 8550.0   -49.65%  | 90917.0   -14.59%        | 7739.0   -54.42% |  |

**Table C1.** Comparison of distance methods and thresholds. Shows the amount (count) of scores per level and percentage difference compared to the original scores. The original scores show the score percentages instead. Considers all nonzero scores of healthy and unhealthy zones (as defined in Sect. 3.3.6) of all turbines with known GEN or GBX failures. \*Note that the totals are obtained by summing  $count \cdot |level|$  and the percentage difference is between these totals, unlike the count used for the percentage difference per level.

*Author contributions*. IV, XC, and TV did the conceptualization and developed the methodology; IV and XC did the data curation, formal analysis, and wrote the paper; IV, XC, TV, JH and AN reviewed and edited the paper; JH and AN acquired funding

Competing interests. The contact author has declared that none of the authors has any competing interests.

Acknowledgements. The authors would like to acknowledge the Energy Transition Funds for their support through the POSEIDON project. This research was supported by funding from the Flemish Government under the "Onderzoeksprogramma Artificiële Intelligentie (AI) Vlaanderen" program and under the VLAIO Supersized 5.0 ICON project. This work used large language model software for spelling, grammar, and sentence construction checks.

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
