# Peer review of "Scalable SCADA-driven Failure Prediction for Offshore Wind Turbines Using Autoencoder-Based NBM and Fleet-Median Filtering"

_Wind Energy Science, 2025_

## Author Response (AR1)

**Author's response**

We are grateful for the thorough feedback we received and will respond to each raised point of each comment separately, and note what we did to address it and what changes we made to the manuscript.

**Comment 1 (RC1)**

1. The filtering method is described as novel, however similar fleet based anomaly filtering strategies have been discussed in prior work (Hendrickx et al. 2020, Li et al. 2020). A clearer articulation of what distinguishes this work is needed.

**Response:** We have adjusted the Introduction and Related Work sections to clarify what makes our filtering method novel and how it differs exactly from the prior work by Hendrickx et al. 2020, Li et al. 2020.

2. The fleet median filtering method assumes most turbines operate under the same conditions at any given time. This assumption may break down, when turbines are shut down for maintenance. Furthermore, in region I downstream turbines produce less power due to wake losses, hence their generator and gearbox temperatures are lower than those of upstream turbines. The authors should discuss how such conditions might affect the effectiveness of the filtering method.

**Response:** We have clarified in the Anomaly Filtering Methodology section that the chance of the fleet median being impacted by turbine shutdowns is negligible by referencing additional papers investigating failure rates and availability of offshore wind turbines. Furthermore, in the same section we added discussion exploring the impact it would have if it did occur. We similarly explain the effect of individual operation patterns, caused by wake or not, on the effectiveness of the filtering method, and why our method does not directly account for it. Then we also clarify that this has been accounted for in our results.

- 3. The scalability of the pipeline is asserted and architecturally supported, but not empirically demonstrated in the manuscript. If this is claimed as a major contribution, the authors should have included for example:
  - Report runtime performance under different fleet sizes
  - · Demonstrate linear or sublinear scaling
  - Show cost, memory or latency metrics as functions of load

**Response:** We have added the Framework subsection in the Result section, describing the deployments of the pipeline framework, detailing and reporting the available metrics, and demonstrating the scaling capability of the used framework features.

**Comment 2 (RC2)**

1. The term "physics-informed" used to describe the filtering method could benefit from further clarification. The description of the filtering method in section 3.3 (distance to fleet median, windowing, multidimensional distances) appears to be primarily statistical and temporal, rather than directly incorporating physical models or principles. It would enhance clarity if the authors could explicitly detail how "physics-informed" aspects are integrated into the filtering logic.

**Response:** Thank you for pointing this out; this term is insufficiently supported, and we have removed it from our descriptions of the filtering method.

2. The paper describes its cloud-based pipeline, highlighting its modularity and scalability for managing anomaly detection across wind farms. However, the contribution of this solution remains unclear as the results section focuses solely on the autoencoder and filtering methods. There are no empirical data or quantitative metrics presented to validate the pipeline's actual performance, scalability, or efficiency.

**Response:** We have added the Framework subsection in the Result section, describing the deployments of the pipeline framework, detailing and reporting the available metrics, and demonstrating the scaling capability of the used framework features.

3. There is not enough detail about the specific failure types examined in this work. The authors mention gearbox and generator failures, but more information is needed about the failure sub-types and their locations for enhanced clarity.

**Response:** We have extended the Turbine data section in the results with all non-confidential information about the failure types that we show in the paper.

4. While the paper acknowledges that data can differ greatly across the fleet and emphasizes the importance of having a large enough fleet for reliable median calculation, I think more discussion is needed about specific sources of variability that could affect the fleet median approach. The paper assumes that a large fleet size will normalize variations, but factors like seasonal variation, turbine location within the wind farm (wake effects, wind exposure differences), and individual operational patterns might create systematic rather than random variations. It would be helpful to have more analysis of how these location-based and operational differences are distinguished from actual anomalies, especially since some turbines might consistently operate differently due to their position rather than equipment issues.

**Response:** We have further clarified in the anomaly filtering methodology section that fleetwide events such as weather anomalies and seasonal variation are automatically accounted for by our method. We also elaborate there on how the effect of individual operation patterns, caused by wake or turbine quirks, affects our method. And how this has been accounted for in our results.

5. Figures 5-7 show negative reconstruction errors and Figures 12-14 show negative anomaly scores. Since reconstruction errors are typically positive differences between predicted and actual values, it's unclear how to interpret negative values in this anomaly detection context.

**Response:** We added a paragraph that clarifies this point to the manuscript. Positive differences are when the observed temperatures are larger than the predicted temperatures. Negative differences are the opposite. Distinguishing the two situations is more informative. For more details, see section 3.1.3 in the manuscript.

6. I would recommend comparing the autoencoder model with other normal behavior modeling approaches, especially since a cloud-based solution has been provided in this work and deployment of autoencoder models could be expensively high. Alternative models like isolation forest, one-class SVM, or statistical approaches might offer better cost-effectiveness and computational efficiency for cloud deployment while achieving similar anomaly detection performance.

**Response:** In section 3.1.3, we have clarified why we used the autoencoder-based NBM and referred to previous research comparing the performance of various NBM implementations. We also included a brief comparison of the computational cost of running the autoencoder model in the new Framework result section, where we discuss the pipeline framework results.